# Protocol for a systematic review of long-term physical sequelae and financial burden of multidrug-resistant and extensively drug-resistant tuberculosis

**Temesgen Yihunie Akalu** [1,2,3]*, **Archie C. A. Clements**[2,4], **Adhanom Gebreegziabher Baraki**[3], **Kefyalew Addis Alene**[1,2,3]

**1** School of Population Health, Faculty of Health Sciences, Curtin University, Perth, Western Australia, Australia, **2** Geospatial and Tuberculosis Research Team, Telethon Kids Institute, Perth, Western Australia, Australia, **3** Institute of Public Health, College of Medicine and Health Sciences, University of Gondar, Gondar, Ethiopia, **4** Peninsula Medical School, University of Plymouth, Plymouth, United Kingdom

* t.akalu@postgrad.curtin.edu.au

**Data Availability Statement:** All the included studies will be fully available online without restrictions when the study is completed and published.

## Abstract

### Introduction

Multidrug resistant (MDR) and extensively drug-resistant (XDR) tuberculosis (TB) are major public health threats that are significant causes of physical sequelae and financial consequences for infected people. Treatment for MDR- and XDR-TB are more toxic and take longer duration than for drug-susceptible-TB. As a result, the long-term sequelae are thought to be more common among patients with MDR- and XDR-TB than drug-susceptible-TB, but this is yet to be quantified. Hence, the aim of this systematic review and meta-analysis is to quantify the global burden and types of long-term physical sequelae and financial burden associated with both MDR- and XDR-TB.

### Method and analysis

We will search CINHAL, MEDLINE, Embase, Scopus, and Web of science for studies that report physical and financial sequelae associated with rifampicin-resistant (RR), MDR- and XDR-TB or their treatments. The search will be conducted without time, language, and place restrictions. A random-effects meta-analysis will be conducted to estimate the pooled prevalence of each physical sequela. Heterogeneity will be measured using the Higgins $I^2$ statistics. We will assess publication bias visually using the funnel plot and statistically using Egger's test. Adjustments for publication basis will be made using Tweedie's and Duval Trim and Fill analysis.

### Ethics and dissemination

Since the study is based on published evidence, ethics approval is not required. The findings of the systematic review will be presented at various conferences and will be published in a peer-reviewed journal.

**Funding:** KAA is funded by the Australian National Health and Medical Research Council (NHMRC) through an Emerging Leadership Investigator Grant APP1196549. TYA is also supported by Curtin University Higher Degree Research (HDR) Scholarship. The funders had no role in the study design, decision to publish, or preparation of the manuscript. Web site for Australian National Health and Medical Research Council (NHMRC): https://www.nhmrc.gov.au/ Web site for Curtin University Higher Degree Research (HDR) Scholarship: https://scholarships.curtin.edu.au/hdr-scholarships-funding/curtin-strategic-scholarships.

**Competing interests:** Authors declare that they have no conflicts of interest.

## Protocol registration

The protocol is published in the PROSPERO with registration number CRD42021250909.

## Introduction

Tuberculosis (TB) is a major cause of death and disability worldwide [1]. According to the recent World Health Organization (WHO) report, about 10.6 million people were ill and 1.6 million people died due to TB in 2021 [1]. South-East Asia (45%), Africa (23%), and Western Pacific regions (18%) were the most affected regions by TB [1]. More recently, the emergence of multidrug-resistant tuberculosis (MDR-TB), resistant to at least isoniazid and rifampicin, and extensively drug-resistant tuberculosis (XDR-TB), MDR-TB plus resistance to any fluoroquinolone and to injectable second-line drugs [1, 2], continue to be a major challenge to achieving the global End-TB Strategy targets [3, 4].

MDR- and XDR-TB are debilitating diseases that can cause serious and long-term physical [5, 6], mental [7], and financial sequelae [8–10] because of the disease process itself or due to side effects related to second-line TB medications [11]. Even though most countries are providing MDR- and XDR-TB diagnosis and treatment free of charge within public health services, there is also a financial crisis because of loss of job due to the disease or due to costs related to second-line medications to treat MDR- and XDR-TB [12].

As the treatment of MDR- and XDR-TB takes longer duration than for drug-susceptible-TB (nine months to two years, and six months, respectively) and is more toxic than for drug-susceptible-TB, the physical sequelae and financial burden are thought to be higher in patients with MDR- and XDR-TB than in patients with drug-susceptible-TB. However, this has not been empirically quantified despite such information being crucial to inform service provision and policymaking in countries where MDR- and XDR-TB are common. Therefore, this systematic review aims to quantify the burden and identify the types of long-term physical sequelae and financial problems associated with MDR- and XDR-TB.

## Materials and methods

This protocol is reported by Preferred Reporting Items for Systematic Review and Meta-Analysis Protocols (PRISMA-P) guidelines [13].

### Search strategy

A systematic search will be conducted to identify potential studies reporting at least one of the primary outcomes (i.e., physical sequelae or financial burden). The systematic search will be conducted on CINAHL(EBSCO), MEDLINE (via Ovid), PsycINFO, PubMed, and Web of Science for relevant studies from the date of each database inception to July 2022. Moreover, the Cochrane Central Register of Controlled Trials (CENTRAL) database will be used for experimental and quasi-experimental studies. Moreover, Gray literature sources like Medicine and drug agencies will be used. To achieve a comprehensive search, the initial search will be supplemented by a PubMed similarity search and a backward and forward citation search of the relevant studies identified as part of our initial search and screening process. Reference lists of included papers will be also screened for additional studies. Corresponding authors will be contacted by email in some instances when additional information is required. The proposed search strategy in PubMed is provided in **Table 1**.

**Table 1. Search strategy.**

| Search | Query |
|---|---|
| #1 | Search "Multidrug-resistan* tuberculosis" OR "multidrug-resistan* TB" OR extensively drug resistan* OR "drug resistan* tuberculosis" OR "drug resistan* TB" OR "Totally drug-resistant tuberculosis" OR"MDR-TB" OR "XDR-TB" OR "Drug-resistant" OR "DR-TB" OR "TDR" |
| #2 | "Blind, Chronic Obstructive Disease, Respiratory Function Tests" OR "Pulmonary Disease, Chronic Obstructive" OR Bronchiectasis" OR "Fibrosis" OR "Disabled Persons" OR "Hearing Loss" OR "Vision Disorders" OR "Heart Failure" OR "Chemical and Drug Induced Liver Injury" OR "Granuloma" OR "Renal Insufficiency" OR "Peripheral Nervous System Diseases" |
| #3 | "Costs" OR "loss" OR "Sales" OR "Loan" OR "Economics" OR "Financ*" OR "Expense" OR "Expenditure*" OR "Payment" OR "Impoverishment" OR "Financial burden" OR "Financial sequelae" OR "Catastrophic health expenditure" OR "Cost of illness" OR "Cost of sickness" OR "Financial stress" OR "Financial stresses" OR "Financial pressure*" OR "Financial hardship" OR "Financial hardships" OR "Financial challenges" OR "Economic burden" OR "Financial toxicity" OR "Financial impact" OR "Health economics" OR "Financial distress". |
| #4 | Search (sequela* "sequels" OR "effect" OR "complication*" OR "illness*" OR "prognos*" OR "function*" OR "chronic*" OR "consequence*" OR "burden*" OR "disabl*" OR "catastroph*" or "impair*") AND ("long-term" or longterm) |
| #5 | **#1 AND (#2 OR #3 OR #4)** |
| #6 | **Search #1 AND (#2 OR #3 OR #4) not animal studies** |

*Truncation

DR-TB: Drug-resistant, MDR-TB: Multi-Drug Resistant Tuberculosis, and XDRTB: Extensively Drug-Resistant Tuberculosis

## Eligibility criteria

**Inclusion criteria.** We will include studies based on the PICO (Population, Intervention, Comparator, and Outcome) question using the following eligibility criteria.

**Participants.** Participants who are reported to be RR, MDR- or XDR-TB patients, from all age groups (i.e., both adults and children) and all regions and countries will be included.

**Intervention.** Our intervention will be a treatment for MDR- or XDR-TB. However, studies without a specific intervention can be also included in this review.

**Comparator or control groups.** Participants who were not diagnosed or treated for MDR- or XDR-TB will be the control groups. This may include patients with drug-susceptible-TB, patients with other diseases, or the general population. Studies without comparison or control groups can be also included in this systematic review.

**Outcome measures.** *Primary outcome measures.* The outcome of the study will be any form of long-term sequelae that have occurred due to MDR-, XDR-TB, or its treatment. The primary outcome of the study will be measured as the proportion of MDR- or XDR-TB patients who developed any form of sequelae, detected, or reported after MDR- or XDR-TB diagnosis, and where MDR- or XDR-TB disease or its medication may have contributed to the sequelae. Long-term sequelae are defined as MDR- or XDR-TB patients who had neurological function impairment, visual impairment (i.e. . . blindness, colour vision impairment), hearing impairment (i.e. . . deafness or hearing loss), musculoskeletal impairment, renal impairment (e.g., renal failure and nephrotoxicity), hepatic failure, cardiac failure, respiratory impairment, and other abnormalities in the lung like fibrosis, chronic obstructive pulmonary disease, bronchiectasis, and lung atrophy. It will also include any long-term sequelae that occurred because of MDR-TB, XDR-TB, or their treatments.

**Types of studies.** In this review, all observational studies including cross-sectional, case-control, and cohort as well as an experimental epidemiological study that reported data on the outcome of interest will be included.

**Exclusion criteria.** Studies will be excluded if they are conference abstracts and animal studies. We will also exclude studies that report treatment outcomes during the treatment period only and those that had insufficient information on the main outcome of interest. Studies conducted only on drug-susceptible TB or latent TB will be excluded.

**Study selection.** Once the literature search has been completed, all identified articles will be imported into an EndNote Library and all the duplicates will be removed. Two independent reviewers (TYA and AGB) will then screen the title and abstract and then review the full text of the papers based on the eligibility criteria. Studies that are not eligible after the screening of titles and abstracts will be excluded. All potentially included studies will be assessed for full-text eligibility by two authors (TYA and AGB). Differences will be resolved by discussion with a third reviewer (KAA). Data from the included studies will be independently extracted by the same two investigators, and information will be collected in a Microsoft Excel (version 2014) spreadsheet.

**Data extraction.** The following data will be extracted from each paper: 1) study characteristics such as name of the first author, year of publication, country of the study, study setting, and study design; 2) participants characteristics such as study populations (i.e. MDR-TB, XDR-TB or both), mean or median age, the proportion of male, sample size; 3) exposure characteristics such as type of MDR- or XDR-TB medications, duration of treatments, comorbidity (HIV, Diabetes Mellitus); 4) outcomes of the study such as type of long-term sequelae, the proportion of people with the sequela, and factors associated with long term sequelae. Where studies use the same dataset, data will be extracted only from the most recent and complete reports. The study will be reported according to the Preferred Reporting Items for Systematic Reviews and Meta-Analysis (PRISMA) statement 2020 (**S1 Appendix**). A completed PRISMA-P checklist is also attached as a (S2 Appendix).

**Quality assessment.** The Newcastle-Ottawa Scale (NOS) will be used to assess the quality of evidence and risk of bias for case-control and cohort studies. The adapted version of the NOS will be used for cross-sectional studies [14]. The Cochrane Handbook for Systematic Reviews of Interventions (Version 6.3, 2022) will be used for assessing the quality of experimental studies [15]. The NOS includes 3 categorical criteria with a maximum score of 9 points, including selection method (4 points), comparability (2 points), and outcome (3 points). Moreover, the Meta-analysis Of Observational Studies in Epidemiology (MOOSE) statement will be used for reporting the results of the systematic review and meta-analysis [16]. Grading of Recommendation Assessment, Development, and Evaluation (GRADE) approach will be used to evaluate the certainty of the evidence. The GRADE approach has five different criteria' (risk of bias, inconsistency, indirectness, imprecision, and publication bias) to evaluate the certainty of evidence [17].

**Data synthesis.** A narrative synthesis will be used to describe the primary outcome of interest. When two or more studies are available, a random-effects meta-analysis will be used to obtain the pooled proportion of patients with each long-term sequela. The funnel plot and Egger's regression test will be used for checking potential publication bias. Trim and Fill analysis will be used as an adjustment method for minor publication bias. Higgins et al method [18] will be used for checking the presence of heterogeneity among included studies and the $I^2$ statistic with its 95% CI will be calculated. A meta-regression analysis will be used to identify potential sources of heterogeneity. A stratified analysis will be conducted WHO geographical region, the income of countries, study design (longitudinal or cross-sectional), quality of the study, and year of publication. The analysis will be conducted using STATA 14 software. A sensitivity analysis will be done by removing low-quality studies.

## Patient and public involvement

Patients will not be involved in the study.

## Ethics and dissemination

Ethical approval is not required for this systematic review as it will be based on a review of published papers. The findings of this review will be published in international journals and presented at international conferences. Short summaries will be widely distributed on social media platforms like Twitter, to reach policymakers and funders.

## Discussion

This systematic review and meta-analysis will comprehensively investigate physical and financial sequelae assoicated with MDR- and XDR-TB. The pooled prevalence of physical sequelae and catastrophic cost among survivors of MDR- and XDR-TB will be measured using a random-effect meta-analysis. The finding will be used as a piece of baseline information for researchers and helpful to planners and decision-makers to minimize the burden of MDR- and XDR-TB sequelae. MDR- and XDR-TB are emerging public health problems causing physical sequelae and financial burdens in affected patients. The type of physical sequelae are varies according to the affected bodily site. For example, people with a history of pulmonary MDR- and XDR-TB often suffer from a range of long-lasting lung complications such as chronic obstructive pulmonary disease, chronic suppurative lung disease (bronchiectasis) [12], chronic empyema [5, 19], pulmonary impairment [20], and life-threatening haemoptysis [21]. MDR- and XDR-TB of the nervous system can affect the meninges, brain [22, 23], spinal cord [24], cranial and peripheral nerves [25] and cause severe irreversible disability such as cerebral hemi-atrophy, paraplegia, and quadriparesis [26, 27]. The treatment of MDR- and XDR-TB requires the use of first- and second-line TB medicines which are associated with adverse events that might also cause long-term sequelae such as visual and hearing impairment [28–30] as well as hepatic and renal dysfunction [31, 32]. This will minimize individual productivity and causes significant economic loss at individual level and population levels.

Understanding the burden of long-term sequelae and associated risk factors and financial burden are crucial to informing policy changes to minimize MDR- and XDR-TB-related poor treatment outcomes and catastrophic costs. This systematic review and meta-analysis will play a vital role in filling the knowledge gap needed to manage MDR-TB and XDR-TB more effectively. While the study will be conducted at a global level without geographic and time restrictions, it will have some possible limitations including significant heterogeneity among studies.

## Conclusion

This systematic review and meta-analysis will be the first to assess the prevalence of physical sequelae and financial burden among patients treated for MDR- and XDR-TB. This study will draw relevant conclusions on common types of physical sequelae, prevalence of sequelae, and burden of catastrophic cost among MDR- and XDR-TB patients, expecting to contribute to achieve the END-TB global target, so to improve the health of MDR- and XDR-TB survivors.

## Supporting information

**S1 Appendix. Preferred Reporting Items for Systematic Reviews and Meta-analyses (PRISMA)-20 flow diagram for the summary of systematic review study selection process.** (DOC)

**S2 Appendix. A completed PRISMA-P checklist.**
(DOCX)

## Author Contributions

**Conceptualization:** Temesgen Yihunie Akalu, Kefyalew Addis Alene.

**Data curation:** Temesgen Yihunie Akalu.

**Formal analysis:** Temesgen Yihunie Akalu.

**Methodology:** Temesgen Yihunie Akalu, Kefyalew Addis Alene.

**Software:** Temesgen Yihunie Akalu.

**Supervision:** Archie C. A. Clements, Kefyalew Addis Alene.

**Validation:** Temesgen Yihunie Akalu, Kefyalew Addis Alene.

**Visualization:** Temesgen Yihunie Akalu.

**Writing – original draft:** Temesgen Yihunie Akalu, Archie C. A. Clements, Adhanom Gebreegziabher Baraki, Kefyalew Addis Alene.

**Writing – review & editing:** Temesgen Yihunie Akalu, Archie C. A. Clements, Adhanom Gebreegziabher Baraki, Kefyalew Addis Alene.

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
