## [Decision Letter · Decision Letter 0]

1 Feb 2023

PONE-D-22-22130Protocol for a systematic review of long-term physical sequelae and financial burden of multidrug resistant and extensively drug resistant tuberculosisPLOS ONE

Dear Dr. Akalu,

Thank you for submitting your manuscript to PLOS ONE. After careful consideration, we feel that it has merit but does not fully meet PLOS ONE’s publication criteria as it currently stands. Therefore, we invite you to submit a revised version of the manuscript that addresses the points raised during the review process.

We look forward to receiving your revised manuscript.

Kind regards,

Jinsoo Min

Academic Editor

PLOS ONE

Journal Requirements:

“No funding”

Additional Editor Comments (if provided):

Please, find the attached reviewer's comments.

PLOS ONE does not copyedit accepted manuscripts, so the language in submitted articles must be clear, correct, and unambiguous. We may reject papers that do not meet these standards.

Reviewers' comments:

Reviewer's Responses to Questions

**Comments to the Author**

1. Does the manuscript provide a valid rationale for the proposed study, with clearly identified and justified research questions?

Reviewer #1: Partly

Reviewer #2: Yes

2. Is the protocol technically sound and planned in a manner that will lead to a meaningful outcome and allow testing the stated hypotheses?

Reviewer #1: Partly

Reviewer #2: Yes

3. Is the methodology feasible and described in sufficient detail to allow the work to be replicable?

Reviewer #1: Yes

Reviewer #2: Yes

4. Have the authors described where all data underlying the findings will be made available when the study is complete?

Reviewer #1: Yes

Reviewer #2: Yes

5. Is the manuscript presented in an intelligible fashion and written in standard English?

Reviewer #1: Yes

Reviewer #2: Yes

6. Review Comments to the Author

You may also provide optional suggestions and comments to authors that they might find helpful in planning their study.

Reviewer #1: This is a systematic review protocol on a very relevant phenomenon in Public Health based on the high impact of tuberculosis and resistance to drugs to treat this disease, to which it must be added that this reality is strongly increased in the countries of the Global South, contributing to increasing social inequalities in health. However, in order to accept this manuscript, I suggest that a number of issues be incorporated:

1. Introduction:

a. Citation number 1 appears to be from the WHO Global TB report, but the 2022 report is available, so the data should be updated. This raises the concern that the numbering of the references does not correspond to the number of the citation in the text, so all citations should be checked to ensure that they correspond to the references. Citation 3 appears before citation 2. The data in the current citation 2 should also be updated with the WHO 2022 report.

b. On lines 54 to 59: Are these the three sentences of reference 4? Even so, they do not seem to correspond because reference 4 alludes to data from an American Indian population.

c. In terms of study objectives, meta-analysis is a statistical technique, not a type of study like a systematic review, which cannot always be carried out, depending on the data found in the selected studies. Therefore, meta-analysis should be removed from the objectives.

2. Methods:

a. A 2015 version of PRISMA is cited, but should be replaced by the latest version in 2020.

b. Within the Search Strategy, Central (trials) from the Cochrane Library and other grey literature sources (Medicines/Drug Agencies). For strategy number 3 (Table 1), it is recommended to introduce more search terms, such as: ("Cost of Illness" OR "Cost of Sickness" OR "Financial Stresses” OR "Financial Stress" OR "financial pressure*" OR "Financial Hardship" OR "Financial Hardships" OR "Financial Challenges" OR "Economic Burden" OR "Financial toxicity" OR "Financial Impact" OR "health economics" OR "Financial Distress").

c. Within the Quality Assessment section, the Cochrane Handbook for Systematic Reviews of Interventions (Version 6.3, 2022) should be incorporated to assess the risk of bias in randomised and non-randomised studies of interventions. In addition, the Newcastle-Ottawa Scale (NOS), the Meta-analysis Of Observational Studies in Epidemiology (MOOSE) statement and the Grading of Recommendation Assessment, Development, and Evaluation (GRADE) need references.

3. Related to Data Availability: As this is a systematic review protocol, no data is currently available in an open access repository, but it would be advisable that the search strategies launched in the different bibliographic databases, as well as the RIS files of the retrieved documents, could be made available once the protocol has been executed.

Reviewer #2: - Line 46-48: I recommend using the recent (2022) global TB report for reporting some figures such as the global and local burden of TB. Almost all the estimates in the introduction should be updated using recent reports.

- In the first search term, I recommend including “Totally drug-resistant tuberculosis” as one of the search terms

- In the third search term, financial sequel, it is better to add “catastrophic health expenditure”, as one of the search terms

- In the fourth search terms, I recommend including terms such as “sequels” and “effect”

7. PLOS authors have the option to publish the peer review history of their article (what does this mean?). If published, this will include your full peer review and any attached files.

Reviewer #1: No

Reviewer #2: No

---

## [Author Response · Author response to Decision Letter 0]

14 Feb 2023

Point-by-point response

Date 4/February/2023

Jinsoo Min

Academic editor

PLOS ONE

Subject: Submission of revised manuscript 

Dear Jinsoo Min, 

Thank you for the opportunity to revise our manuscript “Protocol for a systematic review of long-term physical sequelae and financial burden of multidrug-resistant and extensively drug-resistant tuberculosis”. We would like to thank the reviewers and the editor for their excellent comments which have substantially contributed to improving our manuscript. We have carefully addressed all the comments improving our responses and the revised manuscript accordingly. Below, we have provided point-by-point responses including how and where the text was modified in the main manuscript. Two separate documents, one with track change and the other a clean version are uploaded to the system. We hope that the revised version is now suitable for publication in PLOS ONE and look forward to hearing from you. 

Yours sincerely, 

Temesgen Akalu (on behalf of all authors)

Editor comments Author’s response

Journal Requirements:

 Thank you, editor, for your valuable comments. Based on your comments the manuscript is updated according to the journal guidelines. 

“No funding”

a) Please clarify the sources of funding (financial or material support) for your study. List the grants or organizations that supported your study, including funding received from your institution. KAA is funded by the Australian National Health and Medical Research Council (NHMRC) through an Emerging Leadership Investigator Grant APP1196549. TYA is also supported by Curtin University Higher Degree Research (HDR) Scholarship. We made the change in the online submission and highlighted it in the cover letter.

b) State what role the funders took in the study. If the funders had no role in your study, please state: “The funders had no role in study design, data collection and analysis, decision to publish, or preparation of the manuscript.” The funders had no role in the study design, data collection, analysis, decision to publish, or preparation of the manuscript. We made the change in the online submission and highlighted it in the cover letter.

c) If any authors received a salary from any of your funders, please state which authors and which funders. Temesgen Yihunie Akalu (TYA) is a fully funded Ph.D. student at Curtin University, and he is receiving a fortnightly salary until his Ph.D. course is completed (August/2025). Moreover, Dr. Kefyalew Addis Alene (KAA) is a senior researcher at Curtin University and is funded by the Australian National Health and Medical Research Council (NHMRC). He is also receiving a fortnightly salary at Curtin University. However, the other two reviewers (AGB and ACAC) have no salary at Curtin University. 

d) If you did not receive any funding for this study, please state: “The authors received no specific funding for this work.” Based on your comments the source of funding is updated in a point-by-point response above and the online submission system. It is changed and updated as the study is funded by Curtin University. 

Based on your fruitful comments, the revision was made accordingly. Hence, a caption for supporting information is added to in-text citations and at the end of the manuscript (Page 14, lines 318-319). 

Reviewer 1 Author response

Introduction 

a. Citation number 1 appears to be from the WHO Global TB report, but the 2022 report is available, so the data should be updated. This raises the concern that the numbering of the references does not correspond to the number of the citation in the text, so all citations should be checked to ensure that they correspond to the references. Citation 3 appears before citation 2. The data in the current citation 2 should also be updated with the WHO 2022 report. Thank you, reviewer, for your valuable comments. According to your comment, we used the WHO 2022 global TB report in the introduction section. The change is also highlighted in the manuscript with track change. Citation 2 is also updated and presented before citation 3. You can see the change on page 3 lines 44-60. 

b. On lines 54 to 59: Are these the three sentences of reference 4? Even so, they do not seem to correspond because reference 4 alludes to data from an American Indian population. The citation is updated accordingly in the revised manuscript (Page 3 lines 55 to 60).

c. In terms of study objectives, meta-analysis is a statistical technique, not a type of study like a systematic review, which cannot always be carried out, depending on the data found in the selected studies. Therefore, meta-analysis should be removed from the objectives. Based on your comments the phrase meta-analysis is removed. The change is highlighted on Page 5 lines 92-94.

2. Methods: 

a. A 2015 version of PRISMA is cited, but should be replaced by the latest version in 2020. Thank you, reviewer, for your suggestion. As you said we are planning to use PRISMA-20 for the systematic review and meta-analysis when the review process is completed. However, for protocols, we use PRISMA-P instead of PRISMA. As far as our search is concerned, we couldn’t find PRISMA-20 for protocol. The available PRISMA-20 is not for protocols, but it is for reporting findings of systematic reviews and meta-analyses after the plan is completed. Therefore, the most updated reporting item for systematic review and meta-analysis protocols is the PRISMA-15. Based on recommendations we will use PRISMA -20 for the results once the study is completed. However, we authors are keen to welcome you if you come up with an updated PRISMA-P version 20. 

b. Within the Search Strategy, Central (trials) from the Cochrane Library and other grey literature sources (Medicines/Drug Agencies). For strategy number 3 (Table 1), it is recommended to introduce more search terms, such as: ("Cost of Illness" OR "Cost of Sickness" OR "Financial Stresses” OR "Financial Stress" OR "financial pressure*" OR "Financial Hardship" OR "Financial Hardships" OR "Financial Challenges" OR "Economic Burden" OR "Financial toxicity" OR "Financial Impact" OR "health economics" OR "Financial Distress"). Based on your valuable comments, an additional search from the Cochrane Central Register of Controlled Trials (CENTRAL trial) and Gray literature (Medicines/Drug Agencies) will be made (Page 5 lines 102-105). Moreover, the search term for strategy number 3 is updated accordingly. The change is highlighted in table 1 (searching strategy). 

c. Within the Quality Assessment section, the Cochrane Handbook for Systematic Reviews of Interventions (Version 6.3, 2022) should be incorporated to assess the risk of bias in randomised and non-randomised studies of interventions. In addition, the Newcastle-Ottawa Scale (NOS), the Meta-analysis Of Observational Studies in Epidemiology (MOOSE) statement and the Grading of Recommendation Assessment, Development, and Evaluation (GRADE) need references. It is updated in the revised manuscript and modified as “The Cochrane Handbook for Systematic Reviews of Interventions (Version 6.3, 2022) will be used for assessing the quality of experimental studies”. Besides, we also added references to the Meta-analysis of Observational Studies in Epidemiology (MOOSE) statement and the Grading of Recommendation Assessment, Development, and Evaluation (Grade). The change is highlighted on pages 9-10 lines 173-183. 

3. Related to Data Availability: As this is a systematic review protocol, no data is currently available in an open access repository, but it would be advisable that the search strategies launched in the different bibliographic databases, as well as the RIS files of the retrieved documents, could be made available once the protocol has been executed. As you said no data is available in an open-access repository as it is a protocol. This is why we authors didn’t mention the data availability statement in the manuscript. However, once the protocol has been executed the data will be available in RIS files of the retrieved documents, in proposed search strategies of different databases, and references of included studies will be cited in the manuscript. As a result, the data will be easily available online to everyone interested. So, the availability statement for protocols is not applicable at the movement. 

Reviewer 2 Author’s Response

Line 46-48: I recommend using the recent (2022) global TB report for reporting some figures such as the global and local burden of TB. Almost all the estimates in the introduction should be updated using recent reports. Based on your fruitful comment updated world health organization (WHO) 2022 report was used throughout the introduction section. So, the change is highlighted in a manuscript with a track change. It is also highlighted on page 3 lines 44-60. 

- In the first search term, I recommend including “Totally drug-resistant tuberculosis” as one of the search terms Based on your comment the search strategy is updated in the revised manuscript. Hence, totally drug-resistant tuberculosis (TDR) is included in the revised manuscript (Table 1).

- In the third search term, financial sequel, it is better to add “catastrophic health expenditure”, as one of the search terms Similarly, the third search term is updated and the phrase “catastrophic health expenditure” is included in the revised version. The revision is highlighted in (Table 1).

- In the fourth search terms, I recommend including terms such as “sequels” and “effect” The term “sequels” and “effect” are incorporated in the fourth search term based on your recommendations (Table 1).

---

## [Decision Letter · Decision Letter 1]

14 Mar 2023

PONE-D-22-22130R1Protocol for a systematic review of long-term physical sequelae and financial burden of multidrug resistant and extensively drug resistant tuberculosisPLOS ONE

Dear Dr. Akalu,

Thank you for submitting your manuscript to PLOS ONE. After careful consideration, we feel that it has merit but does not fully meet PLOS ONE’s publication criteria as it currently stands. Therefore, we invite you to submit a revised version of the manuscript that addresses the points raised during the review process.

We look forward to receiving your revised manuscript.

Kind regards,

Jinsoo Min

Academic Editor

PLOS ONE

Additional Editor Comments:

Major revisions

(1) Find other reviews' comments and revise accordingly.

(2) Introduction section is too long. Please, modify the introduction section and reduce to two or three paragraphs. Remnant contents could be moved to the discussion section.

(3) Please, summarize the "methods and materials" and insert a new paragraph immediately after the "Discussion" heading.

(4) Write a new paragraph of 3-4 sentences at the end of the manuscript summarizing all the manuscript as a conclusion.

(5) Try to avoid unnecessary use of the abbreviation, which was only used for once or twice.

page 3 line 45 : COVID-19

page 4 line 84 : DS

page 11 line 205 : COPD

(6) Insert the full term when the abbreviation was first used in the manuscript.

page 2 line 30: RR

page 3 line 59 : DR-TB

page 11 line 221 : PRISMA ?? -> this abbreviation was used elsewhere in the manuscript before.

(7) others

page 9 line 164 : change "diabetes" to "Diabetes" (Use capital letter)

page 11 line 215 : Please, delete sub-heading, "strengths and limitations".

(8) Please, check and correct the all the grammatical errors.

page 5 line 93 : aim -> aims

PLOS ONE does not copyedit accepted manuscripts, so the language in submitted articles must be clear, correct, and unambiguous. We may reject papers that do not meet these standards.

If the language of a paper is difficult to understand or includes many errors, we may recommend that authors seek independent editorial help before submitting a revision. These services can be found on the web using search terms like “scientific editing service” or “manuscript editing service.

Reviewers' comments:

Reviewer's Responses to Questions

**Comments to the Author**

1. Does the manuscript provide a valid rationale for the proposed study, with clearly identified and justified research questions?

Reviewer #1: Yes

Reviewer #2: Yes

2. Is the protocol technically sound and planned in a manner that will lead to a meaningful outcome and allow testing the stated hypotheses?

Reviewer #1: Yes

Reviewer #2: Yes

3. Is the methodology feasible and described in sufficient detail to allow the work to be replicable?

Reviewer #1: Yes

Reviewer #2: Yes

4. Have the authors described where all data underlying the findings will be made available when the study is complete?

Reviewer #1: Yes

Reviewer #2: Yes

5. Is the manuscript presented in an intelligible fashion and written in standard English?

Reviewer #1: Yes

Reviewer #2: Yes

6. Review Comments to the Author

You may also provide optional suggestions and comments to authors that they might find helpful in planning their study.

Reviewer #1: Although the authors use PRISMA-P (2015) for the systematic review protocol reporting items, for the supplementary file (S1 Table: Preferred Reporting Items for Systematic Reviews and Meta-analyses (PRISMA) flow diagram for the summary of systematic review study selection process) they should use PRISMA 2020 (flow diagram for new systematic reviews which included searches of databases, registers and other sources (http://www.prisma-statement.org/PRISMAStatement/FlowDiagram))

Reviewer #2: Authors addressed all my comments on the manuscript. Hence, I recommend this protocol to be published.

7. PLOS authors have the option to publish the peer review history of their article (what does this mean?). If published, this will include your full peer review and any attached files.

Reviewer #1: No

Reviewer #2: No

---

## [Author Response · Author response to Decision Letter 1]

30 Mar 2023

Point-by-point response

Date 30/03/2023

Jinsoo Min

Subject: Submission of revised manuscript 

Dear Jinsoo Min, 

Thank you for providing an additional opportunity to revise our manuscript “Protocol for a systematic review of long-term physical sequelae and financial burden of multidrug-resistant and extensively drug-resistant tuberculosis”. We are grateful for the reviewers' and editor’s comments and believe that the revised manuscript is now stronger because of their feedback. We have carefully addressed all the comments provided by the editor and reviewers in our responses and incorporated related changes into the manuscript. Below, we have provided point-by-point responses including how and where the text was modified in the main manuscript. Two separate documents, one with track change and the other a clean version are uploaded to the system. We hope that the revised version is now suitable for publication in PLOS ONE and look forward to hearing from you in due course. 

Yours sincerely, 

Temesgen Akalu (on behalf of all authors)

Editor comments Author response

Introduction section is too long. Please, modify the introduction section and reduce to two or three paragraphs. Remnant contents could be moved to the discussion section. The introduction section is now modified and reduced into three paragraphs (pages 3 to 4, lines 46-67). Some of the remaining sections moved to the discussion section as suggested by the editor on pages 9-10, lines 176-199. 

Please, summarize the "methods and materials" and insert a new paragraph immediately after the "Discussion" heading. Write a new paragraph of 3-4 sentences at the end of the manuscript summarizing all the manuscript as a conclusion. Following the journal requirements and the editor’s suggestion, we have now summarized the methods and material sections and provided a brief conclusion after the discussion sections (Page 10, lines 201-205).

Try to avoid unnecessary use of the abbreviation, which was only used for once or twice.

page 3 line 45 : COVID-19

page 4 line 84 : DS

page 11 line 205 : COPD All unnecessary use of abbreviations is now modified in the revised version of the manuscript. 

 Insert the full term when the abbreviation was first used in the manuscript.

page 2 line 30: RR

page 3 line 59 : DR-TB

page 11 line 221 : PRISMA ?? -> this abbreviation was used elsewhere in the manuscript before. All abbreviations are corrected in the revised manuscript.

others

page 9 line 164 : change "diabetes" to "Diabetes" (Use capital letter)

page 11 line 215 : Please, delete sub-heading, "strengths and limitations". 

The sub-heading is removed, and the term “diabetes” is changed to “Diabetes”.

Please, check and correct the all the grammatical errors.

page 5 line 93 : aim -> aims All grammatical errors are now corrected in the revised manuscript. 

Reviewer 1 comment Author response

Reviewer #1: Although the authors use PRISMA-P (2015) for the systematic review protocol reporting items, for the supplementary file (S1 Table: Preferred Reporting Items for Systematic Reviews and Meta-analyses (PRISMA) flow diagram for the summary of systematic review study selection process) they should use PRISMA 2020 (flow diagram for new systematic reviews which included searches of databases, registers and other sources (http://www.prisma-statement.org/PRISMAStatement/FlowDiagram)) Thank you for your valuable comment. In the revised version of the manuscript, PRISMA-P (2015) is changed by PRISMA-P (2020).

---

## [Editor Report · Decision Letter 2]

24 Apr 2023

Protocol for a systematic review of long-term physical sequelae and financial burden of multidrug resistant and extensively drug resistant tuberculosis

PONE-D-22-22130R2

Dear Dr. Akalu,

We’re pleased to inform you that your manuscript has been judged scientifically suitable for publication and will be formally accepted for publication once it meets all outstanding technical requirements.

Kind regards,

Jinsoo Min

Academic Editor

PLOS ONE

Additional Editor Comments (optional):

Dear Dr. Akalu,

Thank you for your hard work on revising the manuscript. I am sorry that process of the review system took a long time.

I wish you great success in your future research.

Best regards,

Jinsoo Min
---

## [Editor Report · Acceptance letter]

7 May 2023

PONE-D-22-22130R2 

Protocol for a systematic review of long-term physical sequelae and financial burden of multidrug-resistant and extensively drug-resistant tuberculosis 

Dear Dr. Akalu:

I'm pleased to inform you that your manuscript has been deemed suitable for publication in PLOS ONE. Congratulations! Your manuscript is now with our production department. 

Kind regards, 

on behalf of

Prof. Jinsoo Min 

Academic Editor

PLOS ONE